# Heavy Metal Immobilization Studies and Enhancement in Geotechnical Properties of Cohesive Soils by EICP Technique

**Arif Ali Baig Moghal [1]**, **Mohammed Abdul Lateef [2]**, **Syed Abu Sayeed Mohammed [2]**, **Munir Ahmad [3]**, **Adel R.A. Usman [3,4]** and **Abdullah Almajed [5,*]**

[1] Department of Civil Engineering, National Institute of Technology, Warangal 506004, Telangana State, India; reach2arif@gmail.com or baig@nitw.ac.in

[2] Department of Civil Engineering, HKBK College of Engineering, Bengaluru 560045, Karnataka, India; abdulmohammed040@gmail.com (M.A.L.); abubms@gmail.com (S.A.S.M.)

[3] Soil Sciences Department, College of Food & Agricultural Sciences, King Saud University, P.O. Box 2460, Riyadh 11451, Saudi Arabia; amunir@ksu.edu.sa (M.A.); adosman@ksu.edu.sa (A.R.A.U.)

[4] Department of Soils and Water, Faculty of Agriculture, Assiut University, Assiut 71526, Egypt

[5] Department of Civil Engineering, College of Engineering, King Saud University, P.O. Box 800, Riyadh 11421, Saudi Arabia

\* Correspondence: alabduallah@ksu.edu.sa



**Featured Application: Adopting enzyme induced calcite precipitation technique to reduce swell, permeability and contaminant remediation of soils.**

**Abstract:** Soil treatment methods to cope with ever-growing demands of construction industry and environmental aspects are always explored for their suitability in different in-situ conditions. Of late, enzyme induced calcite precipitation (EICP) is gaining importance as a reliable technique to improve soil properties and for contaminant remediation scenarios. In the present work, swelling and permeability characteristics of two native Indian cohesive soils (Black and Red) are explored. Experiments on the sorption and desorption of multiple heavy metals (Cd, Ni and Pb) onto these soils were conducted to understand the sorptive response of the heavy metals. To improve the heavy metal retention capacity and enhance swelling and permeability characteristics, the selected soils were treated with different enzyme solutions. The results revealed that EICP technique could immobilize the heavy metals in selected soils to a significant level and reduce the swelling and permeability. This technique is contaminant selective and performance varies with the nature and type of heavy metal used. Citric acid ($C_6H_8O_7$) and ethylene diamine tetra-acetic acid (EDTA) were used as extractants in the present study to study the desorption response of heavy metals for different EICP conditions. The results indicate that calcium carbonate ($CaCO_3$) precipitate deposited in the voids of soil has the innate potential in reducing the permeability of soil up to 47-fold and swelling pressure by 4-fold at the end of 21 days of curing period. Reduction in permeability and swell, following EICP treatment can be maintained with one time rinsing of the treated soil in water to avoid dissolution of precipitated $CaCO_3$. Outcomes of this study have revealed that EICP technique can be adopted on selected native soils to reduce swelling and permeability characteristics followed by enhanced contaminant remediation enabling their potential as excellent landfill liner materials.

**Keywords:** EICP; enzyme solution; lead; nickel; zinc; swelling pressure; permeability

## 1. Introduction

Effect of toxic heavy metals on the environment has become a serious concern to the entire world because of the intrusion of heavy metals in the ecosystem [1]. Heavy metals cannot be decomposed and extirpated from the ecosystem because of their stability and cause atrocious health issues to the aquatic and terrestrial life [2]. Further, soil pollution due to heavy metals has become a grave concern to be addressed [3] since it also poses threat to the food security of the world [4], and it is not just hazardous to living creatures but it is a worldwide environmental issue [5,6]. Activities like plating of metals, mining, textile production, dyeing, manufacturing of batteries etc. cause heavy metal pollution in the environment [7]. The scale to which the heavy metal pollution has increased makes it imperative to find for the remediation methods. Hence heavy metal remediation is achieved by different methods like ion exchange, electrodialysis, nanofiltration, flocculation, chemical precipitation, reverse osmosis, ultrafiltration, coagulation and floatation [8]. Mostly adopted method for heavy metal remediation is adsorption due to its flexibility in operation [9]. One of the methods of heavy metal remediation in the soils is immobilization by solidifying the soil by a binder, which reduces the activity and solubility of heavy metals and even improves the mechanical characteristics of soils [10,11].

Given the present scenario wherein the heavy metal contamination has become a grave concern to be addressed at the earliest, methods like soil washing, chemical precipitation, ion exchange, reverse osmosis, electrochemical treatment, oxidation are employed to reduce the adverse effects of heavy metals in polluted environments [12–15]. remediation of heavy metals in fly ashes deposits is also an important issue to be addressed [16]. Use of ground improvement techniques like compaction, stone columns and mixing soil with stabilizers are applied only in undeveloped sites, whereas the soils on which structures already exist can be developed with novel soil improvement methods [17]. Soil modification by biological process, wherein precipitates of calcium carbonate/calcite ($CaCO_3$) are formed in the soil is a promising technique in improving the strength characteristics, reducing permeability and compression behavior of soil [18,19]. EICP technique is a bio-inspired method developed for soil treatment using urease enzyme wherein hydrolysis of urea takes place leading to the $CaCO_3$ precipitation which binds the soil grains together [20–25]. Calcite precipitated by urease can accommodate in the pores between the soil grains and cement them [18,26–28]. $CaCO_3$ is considered effective in the treatment of contaminated waters and shows good removal efficiency for heavy metals [29] and it is also considered as major bio-mineralization material [30]. The dimension of enzyme particle is 12 nm approximately [31,32], this minute size is advantageous in developing biofilms and precipitation of $CaCO_3$ on the soil grains to avoid bio-clogging. This small size of the enzyme is an advantage over microbially induced calcite precipitation (MICP) [31,33], whereas the size of bacteria ranges from 3 to 0.5 µm [34]. Use of MICP technique in soil treatment needs special care to ensure proper precipitation of $CaCO_3$ like providing a proper environment to offer sufficient oxygen for activation of microbes in a deep soil and sometimes voids in soil happen to be smaller than the bacteria cells posing a hindrance to the MICP process, whereas EICP technique is not impacted by these hindrances [35] and MICP used for precipitation of $CaCO_3$ for soil improvement is a complex phenomenon involving inorganic-organic, abiotic-biological and liquid-solid interactions. Factors such as physiological parameters of bacteria, properties of geomaterials, bacterial concentration and cementing solution influence the efficiency and quality of bonding in MICP [36]. Putra et al. [37] have also reported about the difficulties in rearing bacterium in the soil because of various reasons like substances in high amounts disabling the bacteria in urea hydrolysis. Mechanism of enzymatic activity can also be beneficial in heavy metal remediation [38,39].

In-situ application of the EICP technique has vast innovative possibilities because of its ease of implementation in environmental applications and construction like bioremediation and biocementation [40]. Urease enzyme is extensively found in nature and present in different forms in plants, soils, fungi and mammals [41–44]. Urease used in this study is extracted from jack bean, technically termed as *Canavalia ensiformis* [42]. Use of non-fat milk powder is found to create nucleation sites in soil mass; these nucleation sites facilitate the precipitation of $CaCO_3$ by urease enzyme and

delays the process of precipitation of CaCO$_3$, and; this delay in precipitation is beneficial for precipitate morphology [21].

Soil stabilization is usually carried out using conventional stabilizers like cement, lime, fly ash, fibers to enhance the geotechnical properties [45]. The potential applicability of bio-cementation for various field applications has not been explored [46]. The current study relies on the applicability of EICP method in reducing the permeability of soil and its efficacy in restricting the contaminant mobility [47]. It can effectively reduce the built of water near building pits due to seepage [48]. Apart from permeability, the swelling phenomenon of soils due to varying moisture content(s) is also a major concern for majority of civil engineering applications [49]. In majority of the cases, the swell induced stresses overtake their respective soil bearing capacities leading to catastrophic disasters and economic deficits [50]. Moreover, rapid industrialization has led to the pollution of soil and water resulting in accumulation of high concentration of heavy metals which are known to intrude into the cell cycle of living organisms leading to cancer [51]. In lieu of this, there is an urgent need to look for suitable sustainable remedies which can reduce the toxicity of these heavy metals in soils.

In the present study, soils exhibiting different mineralogy are tested for their sorption capacities and removal efficiencies along with swell and permeability characteristics using EICP method. This sustainable study relying on sorption and desorption, will open new horizons in contaminant remediation.

## 2. Materials and Methods

### 2.1. Soil Samples

Soils taken in this study are from Indian origin, Kaolinite type (Red soil) obtained from Bangalore, HKBK College of Engineering (13°02′14.41″ N, 77°37′11.84″) and Montmorillonite (Black soil) procured from Karnataka state Yadgir district (13°02′14.41″ N, 77°37′11.84″). Soils were collected from 3 m depth and sealed in polythene bags to avoid mixing of unwanted biodegradable matter. Geotechnical properties of these soils were determined according to the relevant ASTM standards. Liquid limit of red and black soils were 30% and 54%; plastic limit values were 17% and 27% and Plasticity index was 13% and 27% respectively. According to the Unified Soil Classification System (USCS), red soil was classified as clay with low plasticity and whereas black soil was classified as clay with high plasticity. Coefficient of permeability (K) of the red soil was $5.30 \times 10^{-7}$ cm/s and that for the black soil was found to be $7.83 \times 10^{-8}$ cm/s. Further, the swell pressure was 117.83 kPa and 167.10 kPa for red and black soils respectively. Natural pH of red and black soils was 5.7 and 8.3 respectively. Background concentration of heavy metals (Cd, Ni and Pb) were checked in both the soils by conducting leaching tests and were found be to below detectable limits.

### 2.2. Contaminants

Nitrate salts of Cadmium (Cd(NO$_3$)$_2$), Nickel (Ni(NO$_3$)$_2$) and Lead (Pb(NO$_3$)$_2$) of analytical grade supplied by Winlab Chemical, Market Harborough, United Kingdom, were used in this study, stock solutions (1000 mg/L) of cadmium (Cd), nickel (Ni) and lead (Pb) were used to prepare the desired dosage of contaminant solutions required for the study. Batch sorption tests according to ASTM D4646—16 [52] were conducted to ascertain the amount of heavy metal sorbed on soil. The sorption tests were carried out on the soils with individual heavy metals (Cd, Ni, and Pb) separately, and combination of heavy metal contaminants that is Cd + Ni, Ni + Pb, Pb + Cd and Ni + Cd + Pb were also tested to understand competitive sorption as well as competitive desorption in the soils with and without treatment. The dosage of the combination of heavy metals for sorption tests were 10, 20, 50, and 100 mg/L and solid to liquid (S/L) ratios of 1:4, 1:10, 1:20, 1:50 and 1:100 were maintained for raw as well as for treated soils to conduct sorption tests. Dosage of heavy metal combinations for desorption tests on the soils was 50 and 100 mg/kg. For desorption studies, 50 g oven-dried raw soil samples were washed with distilled water and taken in a container and combination of contaminant solutions of desired dosage (50, 100 mg/kg) prepared from 1000 mg/kg stock solution were mixed

thoroughly and covered with aluminum foil to avoid cross-contamination. Small holes were made on the aluminum foil to facilitate evaporation of water. The containers were kept on an elevated horizontal surface in a humidity regulated chamber for 40 days to assure through interaction between the soil grains and the heavy metal ions at a constant temperature of 27 °C and humidity levels of 40 to 45%. Similar procedure was employed for heavy metal spiked scenario in the soils treated with urease enzyme.

### 2.3. Urease Enzyme

*Canavalia ensiformis* (jack bean) Type III, powder, 15,000–50,000 units/g solid (urease enzyme) was supplied by Sigma-Aldrich, St. Louis, MO, USA. Urease enzyme is a pure protein and common source of the enzyme derived from Jack bean [41]. Additional ingredients used along with urease for preparing the enzyme solutions were calcium chloride dehydrate ($CaCl_2 \cdot 2H_2O$) as it is one of the most preferred and effective sources for elicitation of $CaCO_3$ precipitation [53]. Urea ($NH_2$-CO-$NH_2$) and non-fat milk powder used in the study were sourced from Winlab Chemical, Market Harborough, United Kingdom.

### 2.4. Enzyme Solutions

The basic enzyme solution was prepared by mixing 1 M Urea, 0.67 M Calcium Chloride ($CaCl_2$) dihydrate and 3 g/L urease enzyme in deionized water before mixing in the soil. 4 g/L of non-fat milk powder was added in another Enzyme Solution having same ingredients as in basic enzyme solution and thirdly a low concentration enzyme solution was prepared by mixing 0.37 M urea, 0.25 M calcium chloride and 0.85 g/L urease enzyme and 4 g/L non-fat milk powder. Henceforth, these enzyme solutions are termed as E1, E2 and E3 respectively. Raw soil samples were oven dried and passed through 425 μ sieve and Enzyme solutions by volume equal to the optimum moisture content by weight of the soil were separately mixed in the soil and manually compacted in rigid cylindrical containers. Compaction was done to assure the voids were minimum in the treated soil. The treated soil samples were covered in polythene covers and cured for 21 days to ensure maximum precipitation of $CaCO_3$ [21].

### 2.5. Sorption Tests

Soil samples were initially mixed with distilled water and shaken for 24 h and desired dosage of heavy metal solution was spiked and shaken for 24 h. The solution was filtered using Whatman 42 ashless filter paper and the amount of heavy metal sorbed is calculated using atomic absorption spectrophotometer (AAS) (PerkinElmer Model A-Analyst 400). To ascertain accuracy and reproducibility, each test was performed on three samples and the average value was taken. The amount of heavy metal sorbed on the soil surface was obtained by calculating the difference between the initial and final mass of heavy metal concentration in the filtrate solution. To achieve the desired concentration of heavy metal, i.e., 10, 20, 50, and 100 mg/L; 50 mL of deionized double distilled water is spiked with 0.5, 1, 2.5 and 5 mL of stock solution (1000 mg/L) respectively. All contaminant solutions were maintained to have a solid to liquid (s/l) ratio of 1:20, addition of 1000 mg/L stock solution varies the s/l ratio to 1:20.5, 1:21, 1:22.5 and 1:25 respectively and this variation in s/l ratio is justified by ASTM D3987 [54]. pH of the test samples was maintained to 5 by using 1M HCL (unit molarity). Soil particles get dispersed and hydrated when shaken for a duration of 24 h in deionized double distilled water, and attain steady state. Contaminants are spiked at this stage to enable sorption onto every possible soil grain. To ensure a constant dilution ratio, stock solution of very high concentration was added in a comparatively lesser volume of the aqueous solution to maintain the desired contaminant concentration, i.e., 100 mg/L. It can be observed that the ratio of s/l ratio increases slightly with a difference of 0.1, this is an acceptable method of maintaining the desired contaminant concentration [55].

Different dilution ratios were maintained to carry out the sorption studies. Dilution ratio is the ratio of the amount of liquid in mL taken for a specific mass of soil in grams to be mixed together to

achieve the desired ratio. The ratios maintained in the study were 1:5, 1:10, 1:20, 1:50 and 1:100 by mixing 5 g of dried soil in deionized distilled water of volumes 25, 50, 100, 250 and 500 mL respectively. Further contaminant concentration was maintained at 100 mg/L, by adding 2.5, 5, 10, 25, and 50 mL of 1000 mg/L stock solution. The contaminant stock solutions were added to the samples using the heavy metal stock solution using a micropipette. The solution was shaken for 24 h at 30 rpm before and after adding contaminants. pH value of each aqueous solution was maintained at 5 before adding contaminants to avoid precipitation of the heavy metals and the final pH values of the solutions were determined soon before testing in AAS. Further the same procedure was followed to spike more than one contaminant simultaneously in the following combinations, i.e., Cd + Ni, Ni + Pb, Pb + Cd and Ni + Cd + Pb.

Amount of contaminant sorbed is expressed as sorption coefficient ($q_e$) (mg/g) using the expression:

$$q_e = \frac{(C_0 - C_e)V}{m} \tag{1}$$

where:

$C_0$ = initial concentration of heavy metal (mg/L),
$C_e$ = sorbed heavy metal on the soil (mg/L),
$m$ = mass (mg) of dry soil, and
$V$ = volume (L) of the solution.

Results obtained from the sorption tests were modeled using the Langmuir and Freundlich Isotherms for the soil surfaces. In Langmuir model, the sorption surface offers a single layer coating for heavy metal ion on the specific surface, whereas, the Freundlich model gives insight for sorption on heterogeneous surface(s). The models can be expressed as follows

Langmuir Isotherm

$$q_e = \frac{V_m K C_e}{1 + K C_e} \tag{2}$$

where:

$q_e$ = sorption coefficient (mg/g),
$V_m$ = capacity of monolayer,
$K$ = equilibrium constant,
$C_e$ = sorbed heavy metal on the soil (mg/L).

For ease in model parameters calculation Equation (2) can be written as:

$$\frac{C_e}{q_e} = \frac{1}{V_m K} + \frac{C_e}{V_m} \tag{3}$$

Freundlich isotherm

$$q_e = K_f C_e^{1/n} \tag{4}$$

where:

$K_f$ = Freundlich constant (mg/g)
$1/n$ = intensity of sorption.

For calculating the model parameters Equation (2) can be expressed as:

$$Log q_e = log K_f + \frac{1}{n} log C_e \tag{5}$$

### 2.6. Desorption Tests

Desorption studies were carried out on the soil samples spiked with combination of contaminants, i.e., Cd + Ni, Ni + Pb, Pb + Cd and Ni + Cd + Pb to understand competitive sorption as well as competitive desorption in the soils with and without treatment. To extract the contaminants from

the soils, acid digestion was conducted by citric acid and ethylene diamine tetra-acetic acid (EDTA) using molar concentrations of 0.1, 0.25 and 0.5. The s/l (solid to liquid) ratio was maintained at 1:20.

## 2.7. Swell Test and Hydraulic Conductivity Tests

Soils treated with enzyme solutions and untreated soils were tested for their permeability by conducting falling head permeability test according to ASTM D5084—16a [56] and to ascertain the swell behavior of the soil swelling test (by using oedometer) as per ASTM D4546—14e1 [57] were conducted. When the precast soil sample after curing was put in the permeameter, the gap between the soil sample and inner wall of the mold was filled with cement slurry to avoid movement of water through this gap.

## 2.8. Measurement of $CaCO_3$ Precipitates

To find the amount of $CaCO_3$ precipitated in the soil, acid digestion test was performed [58]. About 50 g of each tested soil samples were taken and immersed in the 1M HCL to dissolve the precipitated $CaCO_3$ and difference in the weights before and after dissolution were used to calculate the percentage of calcite precipitated. Table 1 shows the percentage of precipitated calcite in the soil samples. It was observed that use of E2 showed comparatively higher precipitation which is further justifying the better performance of treated soils with E2.

**Table 1.** Percentage of $CaCO_3$ precipitated by weight of soil in each treated sample.

| Curing (Days) | RS (Permeability Samples) | | | BS (Swell Test Samples) | | |
|:---:|:---:|:---:|:---:|:---:|:---:|:---:|
| | E1 | E2 | E3 | E1 | E2 | E3 |
| 7 | 1.784 | 1.687 | 1.583 | 1.693 | 1.983 | 1.365 |
| 14 | 1.965 | 2.813 | 1.691 | 1.835 | 2.387 | 1.578 |
| 21 | 2.681 | 2.967 | 1.833 | 2.465 | 2.592 | 1.881 |

## 3. Results

### 3.1. Sorption of Individual Heavy Metals

Sorption coefficients ($q_e$) in mg/g were found for the tests conducted on the treated soils, an understanding of the behavior of heavy metal sorption was obtained with the plots shown in Figure 1. Amount of heavy metal getting sorbed on the untreated soil was comparatively lesser than that of the treated soil. Values of sorption coefficients for 100 mg/L initial concentration of Ni were 1.513 mg/g (pH 6.32) and 7.0838 mg/g (pH 7.63) for raw red soil and red soil treated with E2 respectively. For black soil the sorption coefficients for Ni were maximum for treatment with E2, for raw black soil Ni sorbed to 1.8754 mg/g (pH 6.35) and that after treatment with E2 showed an improvement of 7.6873 mg/g while the solution pH was 7.88. Further the sorption was comparatively less effective for Cd and Pb than Ni, and E2 being the most effective in holding the heavy metals. Sorption of Cd increased to 5.4235 mg/g (pH 6.44) and 5.8519 mg/g (pH 7.09) for red soil and black soil treated with E2 from 1.352 mg/g (pH 6.44) and 2.4523 mg/g (pH 6.11) for untreated red soil and black soil respectively. Pb sorption was 5.6925 mg/g (pH 7.01) and 6.6741 mg/g (pH 8.24) for red soil and black soil respectively treated with E2 and that for untreated red soil and black soil was 1.5476 mg/g and 1.8139 mg/g with pH values 6.87 and 7.01 respectively.

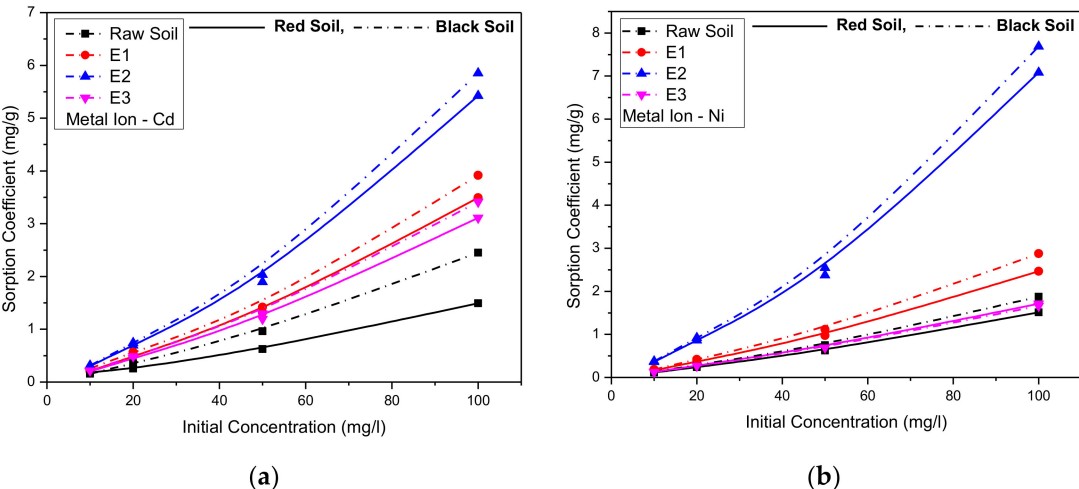

**Figure 1.** Sorption coefficients at different initial concentrations, (**a**) sorption of Cd, (**b**) sorption of Ni.

Sorption on soils treated with E1 and E2 were also better than that of untreated soils, Figure 1a,b show the plots for sorption coefficients against different initial concentrations for both soils spiked with Cd and Ni respectively, similar plot for Pb is given in the Supplementary Materials (Figure S1), it can be observed that sorption for untreated soils was less than that for the treated soils, Thereby indicating the effectiveness of enzyme treated soil in providing suitable environment for the heavy metal to sorb.

The mechanism of increase in sorption can be attributed to the precipitates of $CaCO_3$ on the soil grains which provide sites for heavy metal ions to adsorb [59]. $CaCO_3$ plays a vital role in the improvement of sorption of heavy metals [60]. $CaCO_3$ is known for its involvement in biomineralization of heavy metals like Cu, Ni, Co, Zn, Pb, Ca and Cd and converting them in to precipitates [61], In the present study, precipitated $CaCO_3$ in the soil is the main source to initiate the sorption of heavy metal ions. Adsorption of the heavy metals depends on the parameters like, solubility, adsorbent affinity, size of cations etc. Dixit et al. [51] reported that the heavy metal adsorption takes place due to exchange of protons and precipitation of heavy metals at micro level. Adsorption trend of the heavy metals was Ni > Pb > Cd in both soils. Further the results obtained for sorption tests conducted with different dilution ratios portrayed that, decrease in dilution ratio decreased the sorption coefficient, maximum sorption was observed at a dilution ratio of 1:100 because greater amount of heavy metal will be available to get adsorbed on the soil grain surface and even greater availability of solid surface sites on the soil grains support greater sorption [55]. Figure 2 represents the variation of sorption coefficients for different dilution ratios for Cd and Ni. Sorption coefficients of Pb for different dilution ratios are provided in Supplementary Materials (Figure S2). To understand the distribution of solutes in equilibrium between the different phases, Freundlich and Langmuir models were tested for the adsorption of heavy metals individually spiked in the soils. Details of the Freundlich and Langmuir are provided in the Supplementary Materials (Table S1) and it was observed that the models did not fit to the results obtained from present study.

Sorption coefficients of Cd with 1:100 dilution ratio with concentration of 100 mg/L for untreated red soil was 3.3564 mg/L and 4.711 mg/L for untreated black soils with pH values 5.69 and 5.84 respectively whereas the sorption increased to 6.583 mg/L and 6.6487 mg/L after treating red soil and black soil by E2 and solution pH being 6.29 and 7.32 respectively. Similarly, Ni adsorbed to the maximum extent when compared to Cd and Ni. Table. 1 gives the comparison of sorption coefficients of Cd, Ni and Pb before and after treatment with E2. Comparison with results obtained after E2 treatment is done since E2 is found to provide maximum sorption on the soil.

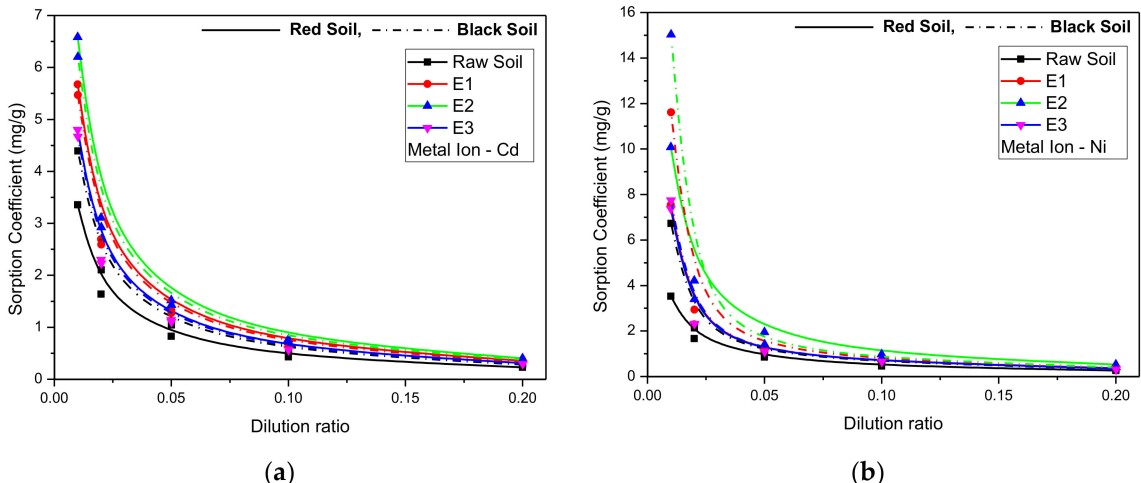

**Figure 2.** Sorption coefficients at different dilution ratios for (**a**) for Cd and (**b**) for Ni.

Impact of pH on the adsorption mechanism is vital, $CaCO_3$ surface is charged positively when pH value happens to be more than 8, even though negative charges exist but are ineffective, thereby attracting the anionic heavy metal content in the solution leading to adsorption [62]. Sorption of Pb on the $CaCO_3$ surfaces is reported to be very high by Godelitsas et al. [63] and they also reported that the mechanism of Pb sorption on the calcite surface also showed the formation of $PbCO_3$ (cerussite), formation of cerussite is helpful in reducing hazardous effects of Pb. Mechanism of sorption of heavy metals on the $CaCO_3$ precipitates is proposed to be ion exchange, precipitation of heavy metal carbonates and hydroxycarbonate forms, adsorption and complexation [64].

Literature available on the use of $CaCO_3$ as a sorbent portrays the use of eggshells [65], precipitates extracted from earthworms [66], naturally available limestone [7], Calcite precipitated by microbes [59], lime [67] it is reported by Makuchowska-Fryc [65] that sorption behavior of heavy metal does not depend on the source of $CaCO_3$ but the initial concentration of the heavy metal in the solution. Hence the mechanism taking place in sorption process by the $CaCO_3$ precipitates by the Enzyme solutions in the present work can be proposed to be similar to that taking place by $CaCO_3$ precipitated by other sources. Ni adsorption was comparatively more than Cd and Pb, the reason for the same may be the presence of Ni in urease used in the work, about 17% of Ni is present in jack bean extracted urease [41], Ni ions already present in $CaCO_3$ precipitated due to the urease enzyme probably club with the Ni ions present in the contaminant solution. It can be observed that the Table 2 shows the statistical analysis of the sorption coefficients for both the soils for different metals and it can be observed that the sorption range for soils treated with E2 is better than that of untreated one.

### 3.2. Simultaneous Sorption of Heavy Metals

Heavy metals spiked simultaneously in the contaminant solution were tested for understanding their sorption behavior. Figure 3 shows variation of sorption coefficient of Cd + Ni and Ni + Pb for different initial concentrations. Similar graphs for Pb + Cd and Ni + Cd + Pb are given in the Supplementary Material (Figure S3). It can be observed from the Figure 3 that simultaneous sorption of heavy metals on raw soils was comparatively less than that for the treated soil and maximum sorption of heavy metals were found taking place on soil treated with E2. Sorption of Cd dominated Ni in raw soil, that is 2.0968 mg/g of Cd and 1.8862 mg/g of Ni was sorbed on the raw red soil sample and the solution pH was 6.21 whereas 4.8215 mg/g and 4.3166 mg/g Cd and Ni respectively were sorbed on red soil treated by E2 while the pH was 7.11 both the results shown here are of initial concentration of 100 mg/L. Sorption coefficients of Cd and Ni for raw black soil were 2.4861 mg/g and 2.4689 mg/g (pH 5.91) respectively and after treatment with E2 sorption increased to 6.8419 mg/g and 5.4178 mg/g (pH 6.14) respectively. Figure 3a,b shows the variation of sorption coefficients for different initial concentrations

for the soil samples treated with E1, E2 and E3 and spiked with Cd + Ni and Ni + Pb respectively, similar graphs for Pb + Cd and Ni + Cd + Pb are shown in Supplementary Materials (Figure S3a,b). Pb adsorption in Figure 3b along Ni is plotted for varying initial concentrations, simultaneous sorption of Ni + Pb on soils also depicted appreciable results after treatment, initially sorption of Ni and Pb were 1.6325 mg/g and 1.2088 mg/g for raw red soil and for black soil 2.8631 mg/g and 2.5611 mg/g respectively at 100mg/L initial concentration, solution pH for raw red and black soils were 5.86 and 5.37 respectively. Sorption coefficient at initial concentration of 100 mg/L for soils treated with E2 were 3.8288 mg/g (Ni) and 4.2109 mg/g (Pb) for red soil and 6.8241 mg/g (Ni) and 5.8057 mg/g (Pb), while the pH values of the contaminant solutions were to 6.33 and 6.21 respectively for red soil and black soil.

**Table 2.** Sorption coefficients of untreated soils and treated with E2 with corresponding SD values.

| Heavy Metal | | Soil | Sorption Coefficient ($q_e$) in mg/L for Raw Soil | ±SD | Sorption Coefficient ($q_e$) in mg/L with E2 Treatment | ±SD |
|---|---|---|---|---|---|---|
| Initial concentration (100 mg/L) | Cd | Red Soil | 1.3520 | | 5.4236 | |
| | | Black Soil | 2.4524 | | 5.8519 | |
| | Ni | Red Soil | 1.5131 | 0.3920 | 7.0838 | 0.8893 |
| | | Black Soil | 1.8754 | | 7.6874 | |
| | Pb | Red Soil | 1.5476 | | 5.6925 | |
| | | Black Soil | 1.8140 | | 6.6741 | |
| Dilution Ratio (1:100) | Cd | Red Soil | 3.3564 | | 6.5830 | |
| | | Black Soil | 4.7110 | | 6.6487 | |
| | Ni | Red Soil | 3.5297 | 1.3352 | 10.0817 | 3.1039 |
| | | Black Soil | 6.7370 | | 15.0400 | |
| | Pb | Red Soil | 3.6910 | | 8.7716 | |
| | | Black Soil | 5.5038 | | 9.4015 | |

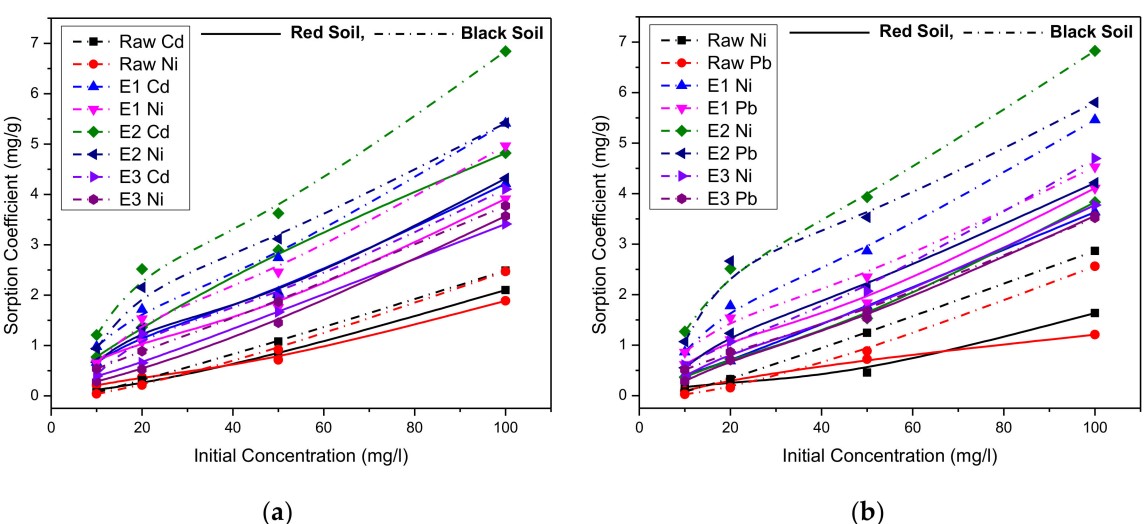

(**a**)                    (**b**)

**Figure 3.** Sorption coefficient of heavy metals spiked simultaneously for different initial concentrations, (**a**) combination of Cd + Ni (**b**) combination of Ni + Pb.

Sorption coefficients for different dilution ratios are plotted in Figure 4 simultaneous sorption of Cd and Ni showed sorption coefficient of 3.7145 mg/g for Cd and that for Ni was 3.5014 mg/g for raw red soil (pH 6.64), whereas sorption of Cd was 5.2036 mg/g and for Ni it was 5.2910 mg/g for raw black soil (pH 6.83), at dilution ratio of 1:100. Sorption of Cd and Ni improved for both soils after treatment

with Enzyme solutions, further E2 happened to be most effective in simultaneous sorption of Cd + Ni. Cd sorption was 7.1624 mg/g and for Ni was 10.6642 mg/g when black soil was treated with E2 and the corresponding pH value was 7.08 and while the sorption of Cd was 9.8116 mg/g and that for Ni was 12.6730 mg/g for black soil treated with E2 and the pH of the contaminant solution was 6.97 at dilution ratio of 1:100. pH is one of the main controlling factors in sorption and solubility of contaminants in the soil [68,69]. Further the sorption of Ni + Pb with different dilution ratios are plotted in Figure 4b. It was observed that E2 treatment gave the best results in sorption of heavy metals whereas treatment with E3 portrayed comparatively lesser sorption than E2 and E1, although sorption after E3 treatment was better than the sorption observed in untreated soil.

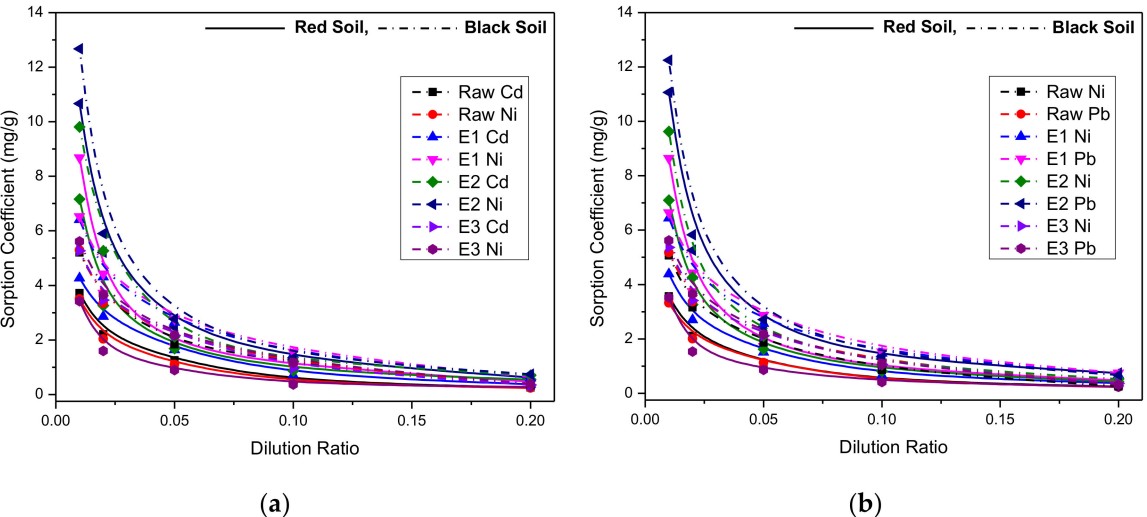

**Figure 4.** Sorption coefficients of heavy metals spiked simultaneously for different dilution ratios: (**a**) combination of Cd + Ni (**b**); combination of Ni + Pb.

Precipitated $CaCO_3$ in the soil has high capability to adsorb metal ions encircling its surface [70–72]. $CaCO_3$ is considered to have large specific surface, hollow structure and steady chemical characteristics, because of which it has found vast application in adsorption of metal ions [73,74] same reasons hold good in this work. Greater adsorption on soils treated with E2 than those treated with E1 and E2 was observed because of higher precipitation of $CaCO_3$ due to the use of non-fat milk powder in E2 [21]. Further electronegativity of metal ions played a vital role in the adsorption process, i.e., metal ions with greater electronegativity get sorbed more than those with lesser electronegativity [75,76]. The electronegativities of the heavy metals used in this study have the order Pb > Ni > Cd that is 2.33, 1.91 and 1.69 respectively. The effect of electronegativity on the sorption process was found justified in present study for different dilution ratios (refer Figure 4a,b) but this reason was not justified in the sorption tests conducted for different initial concentrations (refer Figure 3b). Ionic radii of the heavy metals influence the adsorption process by competing for sites on the adsorbent and metal ions with higher ionic radius induce faster saturation of sites on the adsorbent, therefore the heavy metals move into calcite's crystal lattice and get adsorbed [77], since ionic radius of Pb (202 pm) is greater than that of the other two metals ions (Cd-158 pm and Ni-163 pm) sorption of Pb is comparatively more. Achal et al. [78] studied applicability of bioremediation on Pb contaminated soils by biologically precipitating calcite. The results revealed that Pb changed from soluble/exchangeable (toxic) form to carbonate bound Pb ($PbCO_3$), which is insoluble and nontoxic. Sorption of Cd on the calcite surface is found to be appreciable due to the similarity of its ionic radii with that of Ca ions in the precipitates [79,80]. Makino et al. [81] used $CaCl_2$ for washing Cd contaminated soils and observed 55% removal of Cd from soil. It is attributed to the potential of Cd to form complexes with anions like Cl, $CO_3$, $PO_4$ and $SO_4$ and assisted in the precipitation of $CaCO_3$ to immobilize it even in calcareous soils [30,82]. Results obtained from sorption

studies portrayed that sorption of heavy metals is lesser when compared to that with individual sorption studies probably because of unavailability of sites on the sorbent for the metal ions. This phenomenon is generally found in the sorption processes [83]. A schematic representation of adsorption process has been shown in the Figure 5 accumulation of metal ions encircling the calcite precipitated in the soil pores can be observed there.

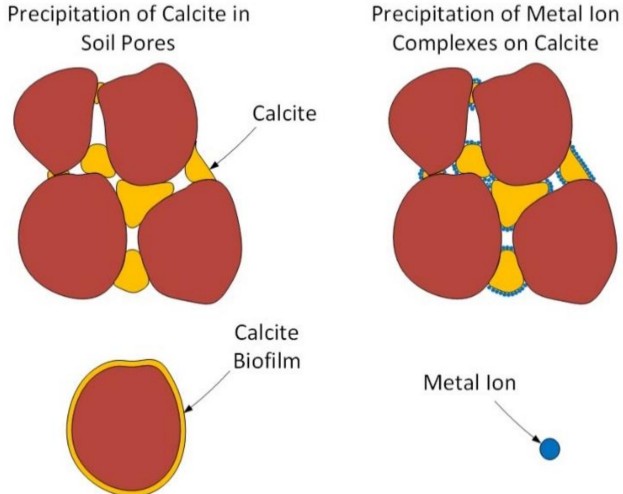

**Figure 5.** Schematic representation of precipitation of calcite in the soil grains and formation of calcite biofilms on the soil grains and adsorption of heavy metal ions on calcite surface.

### 3.3. Simultaneous Desorption of Heavy Metals

Desorption studies give a better understanding of the mechanism of encapsulation since heavy metal solution is introduced in the soil and left for around 40 days to cure.

Figure 6 gives the details variation of removal efficiencies of heavy metals with concentrations 50 and 100 mg/kg for EDTA and citric acid extractants with different molarities. Removal efficiencies of heavy metals in the solution had reduced after treatment with E1, E2 and E3. Figure 6a gives the variation of removal efficiencies for Cd and Ni spiked together with 50 mg/kg load ratio in the soils and extracted by EDTA solution. It was found that removal efficiency of Cd decreased to 11.2% for red soil treated with E2 from 21.28% which was for untreated red soil and removal efficiency of Ni for red soil treated with E2 decreased to 11.64% from 15.24% of untreated red soil. Further for black soil the removal efficiency for Cd and Ni decreased to 11.18% and 10.84% respectively after treatment with E2 from 15.68% and 18.33% for Cd and Ni respectively for untreated black soil. For load ratio of 100 mg/kg of Cd and Ni spiked in the soils and extracted by EDTA the results obtained were again better for E2 treated soils, removal efficiency for Cd and Ni reduced to 10.98% and 6.08% from 12.66% and 10.84% respectively (refer Figure 6c). Similarly, removal efficiencies of Cd and Ni for both load ratios of 50 mg/kg and 100 mg/kg extracted by Citric acid are shown in Figure 6b,d. Similar figures for desorption of other combinations (Ni + Pb, Pb + Cd and Ni + Cd + Pb) are given in Supplementary Materials (Figures S5–S7). $CaCO_3$ precipitated in the soil grains plays vital role in encapsulating the heavy metal heavy metals and method of precipitating $CaCO_3$ by enzymes is specifically effective for Cd [84].

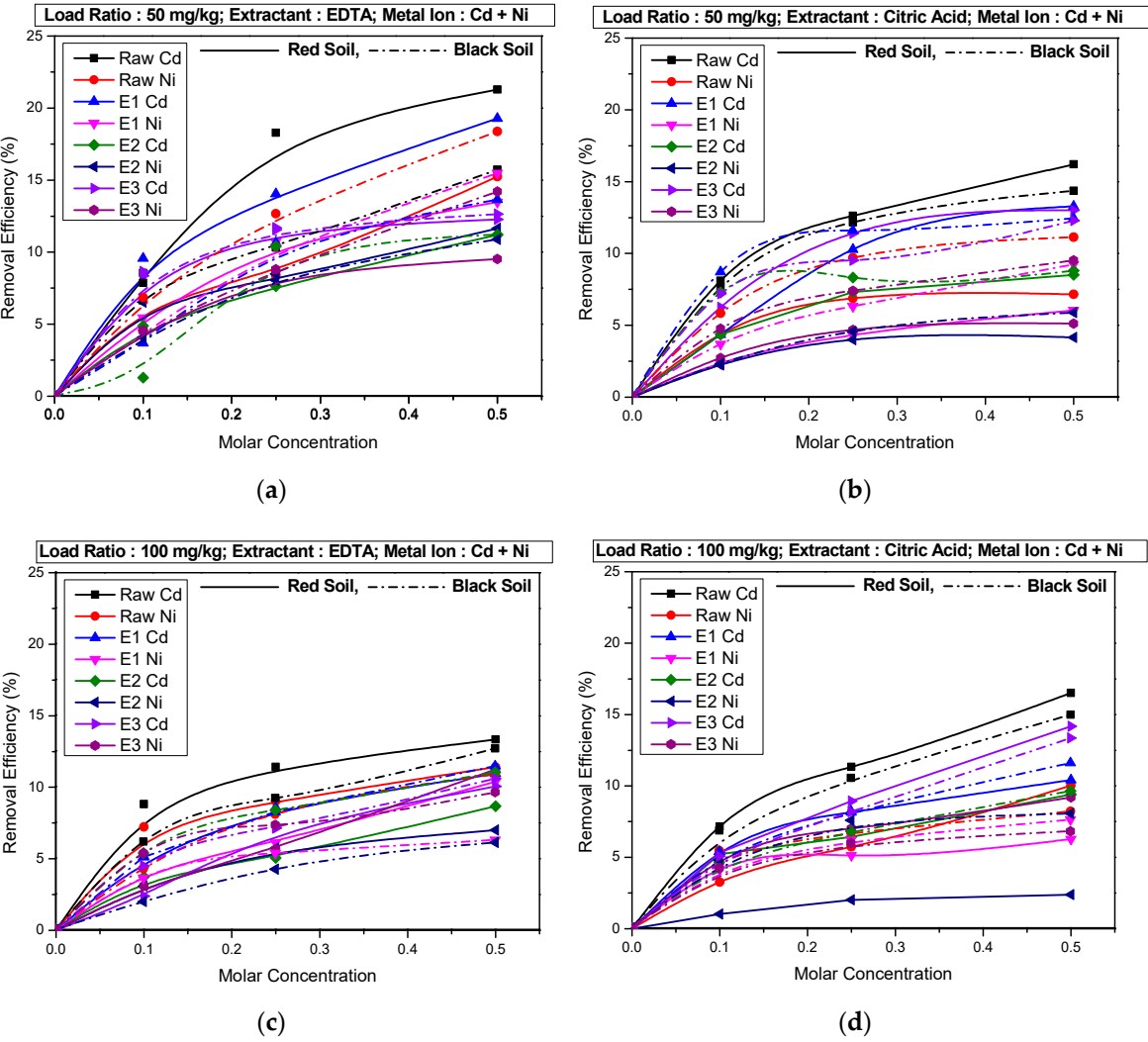

**Figure 6.** Removal Efficiencies of Cd + Ni spiked in both soils with: (**a**) 50 mg/L ethylene diamine tetra-acetic acid (EDTA) extractant; (**b**) 50 mg/L citric acid extractant (**c**); 100 mg/L EDTA extractant, (**d**) 100 mg/L citric acid extractant.

Desorption Studies conducted on all combinations showed that simultaneous immobilization of heavy metals in the soil is better after its treatment, E2 was the most effective solution in decreasing removal efficiencies. It is being noted by Han et al [85] that removal efficiency can be up an average of 80% and removal can be up to 99% with increase in heavy metal concentration thereby posing severe effect on the environment. Continuous accumulation of heavy metal ions from the contamination source leads to serious effects. Applying immobilization techniques on the contaminated sites can reduce the effects by either crystalizing or encapsulating the contaminants. Precipitates of $CaCO_3$ in the soils have different behavior to different heavy metals [77]. The formation of carbonates ($CdCO_3$, $NiCO_3$ and $PbCO_3$) in the form of bio-minerals in contaminated soils was observed by Li et al. [86]. These bio-minerals have different morphologies and include spherical shape, needle shape or rhombohedral shape and minimum dimension of 10 μm to a maximum of 50 μm. Govarthanan et al. [87] carried out lead bio-mineralization in mine tailings using Bacillus sp. KK1 bacteria and found 26% reduction in exchangeable Pb due to the bio-mineralization process triggered by the precipitated calcite.

### 3.4. Permeability and Swell Behavior of Soils

It was observed from the results obtained that the coefficient of permeability (K) of the untreated red soil was $5.3 \times 10^{-7}$ cm/s and for red soil treated with E2, K reduced to $1.12 \times 10^{-10}$ cm/s after a curing period of 21 days. Whereas black soil also portrayed appreciable reduction in K. Untreated black soil had K = $7.83 \times 10^{-8}$ cm/s and the same reduced to $6.31 \times 10^{-11}$ cm/s. This reduction in permeability is probably due to the adhesion of soil grains due to the precipitated $CaCO_3$ in the voids of the soil mass, making the movement of water through the existing pore spaces, which create a longer path [88]. It can be observed in Figure 7a that irrespective of the enzyme solutions used to treat the soils there is a minimum reduction of K = $5.2884 \times 10^{-7}$ cm/s for red soil and K = $7.77779 \times 10^{-8}$ for black soil and Figure 7b shows the variation of K for different curing periods for treated and untreated soils tested by contaminant (Ni + Cd + Pb) induced solution and the results obtained are almost nearer to that observed in the Figure 7a. Therefore, it can be inferred that percolation of contaminated fluid in the soil mass doesn't affect improved behavior of soil permeability after treatment by enzyme solutions. Nemati and Voordouw [89] studied the effect on permeability of porous media by enzymatic $CaCO_3$ precipitation and found that precipitates successfully helped in reducing the permeability and EICP method depended on the concentration of urease and urea and calcium chloride reactants. Therefore, in this study it can be proposed that the $CaCO_3$ precipitates help in reducing the permeability of the treated soils as it can be observed from the Figure 7a. Further use of EICP to plugging of pores to reduce the permeability is also a promising and efficient method which can be adopted on porous media [90,91]. It is also evident that $CaCO_3$ precipitates are effective in giving highest strength and minimum permeability [92], because it hardens in the soil pores [93]. E2 gave the best results in reducing the permeability of the soil which can be due to the use of not-fat milk powder. Non-fat milk powder facilitates the creation of nucleation sites in the soil mass paving the way to the precipitation of $CaCO_3$ in the soil mass stably [21]. $CaCO_3$ clusters formed in the soils by enzyme treatment also influence the permeability of soils in comparison with MICP, value of K happens to be more for enzyme treated soils [94] this scenario is probably due to use of enzyme without non-fat milk powder where distribution of nucleation cites in the soil mass doesn't take place uniformly. Reduction in permeability after enzyme treatment of porous media has been observed to reach up to 98% by Nemati et al. [89].

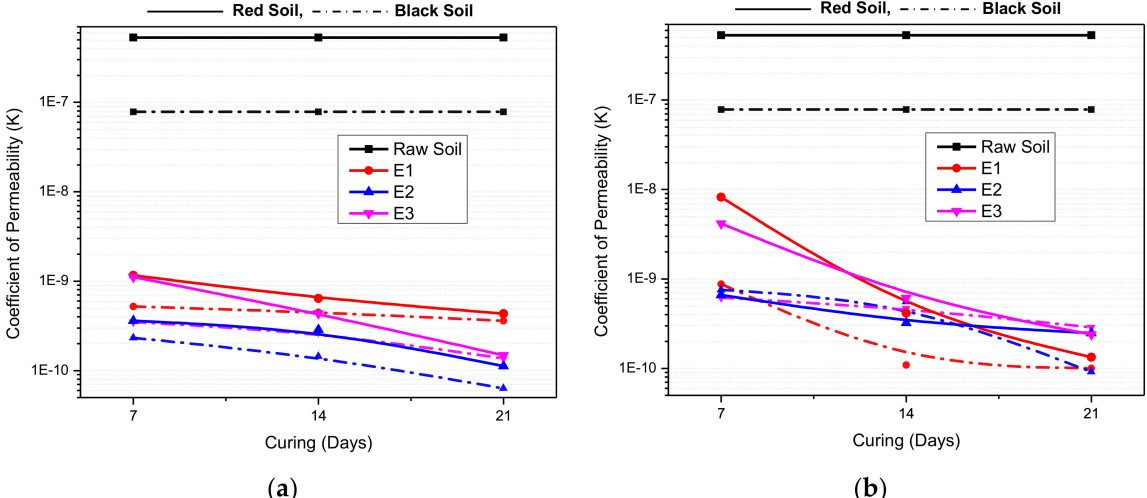

**Figure 7.** Variation of K for both soils for curing period of 7, 14 and 21 days for treated and untreated soil: (**a**) for plain water; (**b**) for contaminant (Ni + Cd + Pb) solution.

Swell test results obtained from the present study are plotted in the Figure 8. it is observed that the swell pressure reduced with treatment of enzyme solutions and age of specimen. It can be observed that E2 gave the best results for both soils, the swell pressure for the untreated red soil was

found to be 117.83 kPa and that for the untreated black soil was 167.1 kPa and after enzyme treatment the swell pressure reduced to 37.68 and 47.2 kPa respectively for soils treated with E2 and cured for 21 days. Control of swell characteristics of expansive soils can be achieved by precipitated $CaCO_3$ [95]. This improvement in the swell characteristics may be due to adhesion of soil grains to each other after $CaCO_3$ precipitation. Further EICP technique which is effective in improving the UCS values [96] of the treated soil which is an indication that the externally applied load seems less effective on the soil because of the $CaCO_3$ precipitates, same reason can be attributed to the reduction of the swell pressure in the soil wherein, the capillary water seems to detach the soil grains. Possibility of failure can also be attributed to particle breakage, since precipitation of $CaCO_3$ in the pores of soil grains increases the chances of soil grains getting crushed because of externally applied load decreases, and particle breakage is also reduced because of increase of diameter of soil grain and precipitates as a whole leading to increase in the capability of withstanding load.

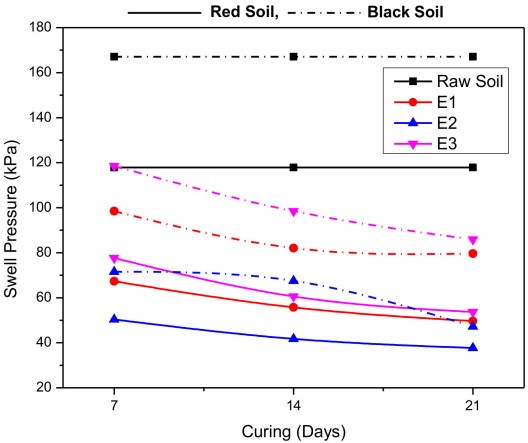

**Figure 8.** Variation of swell pressure of in the soils before and after treatment.

Further $CaCO_3$ acting as bonds between the soil grains continently accommodates itself to dissipate loads thereby avoiding formation of cleavage between soil grains. Lastly even if $CaCO_3$ gets detached from the soil grains due to excess loads still offers cushioning effect to the adjacent soil grains to effectively form chain of forces in the soil mass to transfer to nearest hard surface [97]. Therefore, due to the adhesion of the soil grains by precipitated $CaCO_3$ swell pressure of the soil can be reduced.

The improvement in stiffness and strength of the soil is attributed to the dense nature of soil grains which create maximum sites of nucleation for $CaCO_3$ precipitation enabling the load transfer between the individual soil grains through inter-particle contacts [98]. These inter-particle bonds also serve the purpose of blocking the percolation of water and cause net reduction in the permeability. Furthermore, it also reduces the effect of capillary water surface tension in the soil mass which contributes to the development of swell pressure.

The mechanism of $CaCO_3$ precipitation takes place by the hydrolysis of urea ($CH_4N_2O$) where in the urea when combined with water is decomposed to form carbonate ions and ammonium. These carbonate ions in the presence of calcium ions (fetched by the addition of $CaCl_2$) in aqueous solution gets a suitable environment because of increase in pH due to the formation of ammonia, and favors faster precipitation of $CaCO_3$ [98]. In addition, the use of non-fat milk powder in E2 which has shown comparatively better results than E1 and E3 has a peculiar phenomenon of precipitation of $CaCO_3$. Casein present in milk products act as an agent creating the nucleation sites in the soil and advocates the precipitation of $CaCO_3$ and its growth, due to which the precipitates holding the soil grains together become stable and the use of milk product also slows down the precipitation rate which may lead to precipitation of large crystals of calcite [21]. $CaCO_3$ crystals which are precipitated in the soil mass have different forms which are named as calcite, vaterite and aragonite, these forms

have same chemical formula but the crystal structure differ [99]. Among the three types of precipitates observed by the researchers, calcite is considered as one of the most stable form having rhombohedral structure [100–102]. EICP technique adopted for soil treatment in this work has $CaCO_3$ precipitation in the form of calcite hence the performance of the treated soil is better when compared to the untreated one.

## 4. Conclusions

Soil improvement relying on enzymes is found promising in enhancing the strength and stiffness properties of cohesive clays while improving their adsorption abilities. Efficacy of EICP method on adsorption and desorption of soils spiked with combination of heavy metals and for individual case(s) gave encouraging results. The following conclusions are drawn from the study:

- Heavy metal adsorption of soils improved followed by enzyme treatment with E2 outperforming E1 and E3. When heavy metal ions were spiked in the soils individually the order of sorption was Ni > Pb > Cd for red soil and black soil at initial concentration of 100 mg/L and at dilution ratio of 1:100. Ni sorption increased by seven-fold at an initial concentration of 100 mg/L. Further at a dilution ratio of 1:100, Ni sorption was found to increase 10 manifolds for red soil and 15 manifolds greater for black soil with enzyme treatment.
- When heavy metals were simultaneously spiked in the soils, competition of heavy metal sorption was observed and it portrayed that heavy metals with higher ionic radius and electronegativity exhibited greater affinity in occupying adsorbent surface. Black soil exhibited comparatively greater ability in sorbing metal ions onto its surface compared to red soil.
- Desorption studies on simultaneously spiked heavy metals gave an understanding that a probable formation of heavy metal carbonates ($PbCO_3$, $CdCO_3$ and $NiCO_3$) on the soil surfaces takes place after $CaCO_3$ interacts with the spiked heavy metal ions as the pH value increases. These arbonates are insoluble and reduce the toxicity of heavy metals.
- Marked reduction in permeability values was observed with EICP treatment and this phenomenon was more pronounced with E2. Non-fat milk powder in E2 facilitates effective $CaCO_3$ precipitation due to the presence of casein protein causing this reduction in permeability. For red soil, permeability values reduced from $5.3 \times 10^{-10}$ cm/s to $1.12 \times 10^{-10}$ cm/s with E2 treatment at 21 days of curing period. For black soil, permeability reduced from $7.83 \times 10^{-11}$ cm/s to $6.31 \times 10^{-11}$ cm/s with E2 treatment.
- Swell characteristics significantly improved upon enzyme treatment. The reduction in swell pressure was three-folds with red soil and three and a half folds with black soil with E2 treatment 21 days of curing period. Calcite precipitates at nucleation sites cause improved contact points between soil grains enabling them to withstand the swell pressures.

EICP method has proven effective in addressing the problems posed by expansive soils and for remediation of heavy metals. This promising method can be opted for in-situ applications to reduce the toxicity levels of brownfields. EICP method is an environment friendly, sustainable, high on green ratings, low on carbon emissions and practical for field applications and can be a promising substitute for conventional mechanical stabilization techniques.

**Supplementary Materials:** The following are available online at http://www.mdpi.com/2076-3417/10/21/7568/s1, Figure S1: Sorption coefficients of Pb at different initial concentrations, Figure S2: Sorption coefficients of Pb at different dilution ratios, Figure S3: Sorption coefficients of (a) Pb + Cd and (b) Ni + Cd + Pb for different initial concentrations, Figure S4: Sorption coefficients of (a) Pb + Cd and (b) Ni + Cd + Pb for different dilution ratios, Figure S5: Removal Efficiencies of Ni + Pb spiked in both soils with, (a) 50 mg/L EDTA extractant, (b) 50 mg/L Citric acid Extractant (c) 100 mg/L EDTA extractant, (d) 100 mg/L Citric acid Extractant, Figure S6: Removal Efficiencies of Pb + Cd spiked in both soils with, (a) 50 mg/L EDTA extractant, (b) 50 mg/L Citric acid Extractant (c) 100 mg/L EDTA extractant, (d) 100 mg/L Citric acid Extractant, Figure S7: Removal Efficiencies of Ni + Cd + Pb spiked in both soils with, (a) 50 mg/L EDTA extractant, (b) 50 mg/L Citric acid Extractant (c) 100 mg/L EDTA extractant, (d) 100 mg/L Citric acid Extractant, Figure S8: Swell test in progress. Table S1: Freundlich and Langmuir model calculations for sorption of individual heavy metals sorbed on soils.

**Author Contributions:** Experimentation and drafting of this paper is taken up by M.A.L., S.A.S.M. and A.A.B.M.; further the entire work was conceptualized by S.A.S.M. and A.A.B.M.; experimental design and plan for conduction of experiments were by, A.A.B.M., M.A.L., M.A. and S.A.S.M.; funding acquisition for the work and literature was perceived by M.A., A.R.A.U. and A.A.; interpretation of data was made by A.A.B.M., M.A.L., S.A.S.M., A.A. and A.R.A.U. All authors have read and agreed to the published version of the manuscript.

**Funding:** Deputyship of Research & Innovation, Ministry of Education, Saudi Arabia: IFKSURG-1440-073.

**Acknowledgments:** The authors extend their appreciation to the Deputyship of Research & Innovation, Ministry of Education, Saudi Arabia for funding the study through Research Group No: IFKSURG-1440-073.

**Conflicts of Interest:** The authors declare no conflict of interest.

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
