# Peer review of "Heavy Metal Immobilization Studies and Enhancement in Geotechnical Properties of Cohesive Soils by EICP Technique"

_applsci, doi:10.3390/app10217568_

Round 1

Reviewer 1 Report

Research Significance: Please add more at the end of the introduction.

Results: Discussions need to be improve by adding some references in the discussion section and a comparison of the findings of previous and other researches should be given. 

Conclusions: Addition of some numerical values in the conclusions would enhance the quality of the study. 

Author Response

Author Response Sheet to Reviewer # 1

Manuscript ID: applsci-984562
Title: Heavy Metal Immobilization Studies and Enhancement in Geotechnical Properties of Cohesive Soils by EICP Technique

The authors thank the reviewer for constructive comments. The authors have addressed all the comments and marked them in the revised manuscript using RED Color Font.

Query 1: Research Significance: Please add more at the end of the introduction.

Response: Agreed. The following content has been added in the revised manuscript (marked in Red Font).

Soil stabilization is usually carried out using conventional stabilizers like cement, lime, fly ash, fibres to enhance the geotechnical properties [45]. The potential applicability of bio-cementation for various field applications has not been explored [46]. The current study relies on the applicability of EICP method in reducing the permeability of soil and its efficacy in restricting the contaminant mobility [47]. It can effectively reduce the built of water near building pits due to seepage [48]. Apart from permeability, the swelling phenomenon of soils due to varying moisture content(s) is also a major concern for majority of civil engineering applications [49]. In majority of the cases, the swell induced stresses overtake their respective soil bearing capacities leading to catastrophic disasters and economic deficits [50]. Moreover, rapid industrialisation has led to the pollution of soil and water resulting in accumulation of high concentration of heavy metals which are known to intrude into the cell cycle of living organisms leading to cancer [51]. In lieu of this, there is an urgent need to look for suitable sustainable remedies which can reduce the toxicity of these heavy metals in soils.

In the present study, soils exhibiting different mineralogy are tested for their sorption capacities and removal efficiencies along with swell and permeability characteristics using EICP method. This sustainable study relying on sorption and desorption, will open new horizons in contaminant remediation. 

Query 2: Results: Discussions need to be improve by adding some references in the discussion section and a comparison of the findings of previous and other researches should be given.

Response: Agreed and revised as suggested. New references are included in the discussion part and the comparisons are made with the findings from research carried out earlier researchers.

Query 3: Conclusions: Addition of some numerical values in the conclusions would enhance the quality of the study. 

Response: Agreed and revised as suggested. Conclusions are updated with numerical values.

--------------------------------------------------------------------------------------------------------------------------------------

Reviewer 2 Report

General comments:

This manuscript examines the sorption capacity and the removal efficiency of heavy metals as well as swelling and permeability characteristics on the MICP-treated soils. The topic is of significant importance for geotechnical engineers. The authors have conducted a number of laboratory experiments, which is praiseworthy. The experimental procedures are well organized and the obtained results are easy to be followed. The efficacy of the precipitated CaCO3 on the encapsulation of heavy metals and the mechanisms of the sorption are well understood. This reviewer enjoyed reading this manuscript.

In conclusion, this reviewer recommends the current work for publication after minor revision. Specific comments are as follows.

Specific comments:

  1. Line 76: Please add a reference for 12 nm of the size of enzyme particle.

  1. Lines 110 and 111: Please check the effective digits of the permeability (5.3 vs 7.83 (2 digits or 3 digits?)) and sell pressure (117.83 vs 167.1 (5 digits or 4 digits?)).

  1. Lines 138 and 139: Please correct the following sentence. It is grammatically wrong. “Urease enzyme is derived from Jack bean is a pure protein and common source of the enzyme”

  1. Line 170: Can the 24-hour shaking reach “equilibrium state”? If the expression of “steady state” is more appropriate, please change it.

  1. Table 1: The definition of the CaCO3 percentage should be clearly described. It is the weight ratio of CaCO3 to soil particles?

  1. Line 316: Please delete a period after Figure3. Likewise, please check Lines 420, 422, 424 for “Figure 7. (a)” and “Figure 7. (b)”, and Line 443 for “Figure. 8”.

  1. Line 357: metals ions => metal ions

  1. Line 454: attribute => attributed

Author Response

Author Response Sheet to Reviewer # 2

Manuscript ID: applsci-984562
Title: Heavy Metal Immobilization Studies and Enhancement in Geotechnical Properties of Cohesive Soils by EICP Technique

The authors thank the reviewer for constructive comments. The authors have addressed all the comments and marked them in the revised manuscript using RED Color Font. The page numbers which have been referred by the reviewer correspond to first revision. In the revised version, due to addition of new sentences as well as comprehensive editing undertaken, the respective line numbers would have changed. The author(s) have responded to the queries in this response sheet relying on the same line numbers.  

Query 1: Line 76: Please add a reference for 12 nm of the size of enzyme particle.

Response: Reference has been added in the revised manuscript. 

References are added to the above-mentioned line in the manuscript.

Query 2: Lines 110 and 111: Please check the effective digits of the permeability (5.3 vs 7.83 (2 digits or 3 digits?)) and sell pressure (117.83 vs 167.1 (5 digits or 4 digits?)).

Response: Corrected. The effective digits are in the revised manuscript.

“Coefficient of permeability (K) of the red soil was 5.30x10-7 cm/s and that for the black soil was found to be 7.83x10-8 cm/s. Further, the swell pressure was 117.83 kPa and 167.10 kPa for red and black soils respectively.”

Query 3: Lines 138 and 139: Please correct the following sentence. It is grammatically wrong. “Urease enzyme is derived from Jack bean is a pure protein and common source of the enzyme”

Response: Apologies. It has been corrected in the revised manuscript. The revised sentence reads as: “Urease enzyme is a pure protein and common source of the enzyme derived from Jack bean”

Query 4: Line 170: Can the 24-hour shaking reach “equilibrium state”? If the expression of “steady state” is more appropriate, please change it.

Response: The authors agree with the reviewer. In fact, expressing the condition as “steady state” is more emphatic. The sentence is redrafted in the revised manuscript as under.

“Soil particles get dispersed and hydrated when shaken for a duration of 24 h in deionised double distilled water, and attain steady state”

Query 5: Table 1: The definition of the CaCO3 percentage should be clearly described. It is the weight ratio of CaCO3 to soil particles?

Response: Agreed and Revised as suggested. CaCO3 is the weight ratio to soil particles. Table 1 has been named as ““Table 1. Percentage of CaCO3 precipitated by weight of soil in each treated sample”

Query 6: Line 316: Please delete a period after Figure3. Likewise, please check Lines 420, 422, 424 for “Figure 7. (a)” and “Figure 7. (b)”, and Line 443 for “Figure. 8”.

Response: Agreed and revised as suggested. The periods are removed in the revised manuscript and these corrections appear at following places.

Line 316: “It can be observed from the Figure 3 that simultaneous sorption”
Lines 420: “It can be observed in Figure 7 (a) that irrespective of the Enzyme Solutions used to treat the soils”
Lines 422: “and Figure 7 (b) shows the variation of K for different curing periods”
Lines 424: “observed in the Figure 7 (a)

Query 7: Line 357: metals ions => metal ions

Response: Correction incorporated in the revised manuscript. The sentence is corrected to:

“because of which it has found vast application in adsorption of metal ions”

Query 8: Line 454: attribute => attributed

Response: Correction incorporated in the revised manuscript. The sentence is corrected to:

 “Possibility of failure can also be attributed to particle breakage”
